# Are bivalves susceptible to domestication selection? Using starvation tolerance to test for potential trait changes in eastern oyster larvae

**Katherine McFarland**[1,2,3]\*, **Louis V. Plough**[2], **Michelle Nguyen**[2,4], **Matthew P. Hare**[1]

**1** Department of Natural Resources, Cornell University, Ithaca, New York, United States of America,
**2** Center for Environmental Science University of Maryland, Cambridge, Maryland, United States of America,
**3** NOAA Fisheries NEFSC, Milford Laboratory, Milford, CT, United States of America, **4** College of Earth, Ocean, and Atmospheric Sciences, Oregon State University, Corvallis, Oregon, United States of America

\* Katherine.m.mcfarland@noaa.gov

**Data Availability Statement:** Data for this study are available at: https://doi.org/10.7298/hbtc-1433.

**Funding:** Funding was provided by the Atkinsons Center for a Sustainable Future (Cornell University)

## Abstract

Conservation efforts are increasingly being challenged by a rapidly changing environment, and for some aquatic species the use of captive rearing or selective breeding is an attractive option. However, captivity itself can impose unintended artificial selection known as domestication selection (adaptation to culture conditions) and is relatively understudied for most marine species. To test for domestication selection in marine bivalves, we focused on a fitness-related trait (larval starvation resistance) that could be altered under artificial selection. Using larvae produced from a wild population of *Crassostrea virginica* and a selectively bred, disease-resistant line we measured growth and survival during starvation versus standard algal diet conditions. Larvae from both lineages showed a remarkable resilience to food limitation, possibly mediated by an ability to utilize dissolved organic matter for somatic maintenance. Water chemistry analysis showed dissolved organic carbon in filtered tank water to be at concentrations similar to natural river water. We observed that survival in larvae produced from the aquaculture line was significantly lower compared to larvae produced from wild broodstock (8 ± 3% and 21 ± 2%, respectively) near the end of a 10-day period with no food (phytoplankton). All larval cohorts had arrested growth and depressed respiration during the starvation period and took at least two days to recover once food was reintroduced before resuming growth. Respiration rate recovered rapidly and final shell length was similar between the two treatments Phenotypic differences between the wild and aquaculture lines suggest potential differences in the capacity to sustain extended food limitation, but this work requires replication with multiple selection lines and wild populations to make more general inferences about domestication selection. With this contribution we explore the potential for domestication selection in bivalves, discuss the physiological and fitness implications of reduced starvation tolerance, and aim to inspire further research on the topic.

awarded to MPH, the National Science Foundation Research Experience for Undergraduates fund awarded to MN, and the Deerbrook Charitable Trust (DCT# 17-26) awarded to LVP.

**Competing interests:** The authors have declared that no competing interests exist.

## Introduction

For many regions, climate change predictions suggest rising temperatures and dramatic variations in precipitation [1] that will stress estuarine and coastal populations through rapid changes in salinity and the spread and proliferation of disease [2,3]. Given the urgency of conservation needs to maintain population viability under rapid environmental change, the prospects for "assisted evolution" using selective breeding or developmental manipulations have increasingly been discussed and investigated [4]. One concern with selective breeding as part of a population management strategy is that captivity itself can impose unintended artificial selection [5,6]. Evolutionary responses to this "domestication selection" can be swift in captive populations [5,7,8] with the potential for reduced fitness in the wild relative to wild born individuals [9]. The propensity for rapid domestication selection is tied to life history because selection can be especially strong on cohorts of high fecundity organisms with high early mortality (type III survivorship curve). When domestication selection is strong within a single propagation cohort, such as was demonstrated for salmonids [5], then its effects can potentially impact the success of hatchery-based population supplementation. These considerations are of particular relevance to marine bivalves because (1) they express extreme versions of this life history, (2) native bivalve populations are depleted in some areas and receiving hatchery-based population supplementation [10], and (3) selectively bred lines intended for commercial aquaculture have been promoted and used for population supplementation, for example in an attempt to mitigate disease mortality [11]. Unfortunately, there is virtually nothing known about the prevalence of domestication selection in aquatic organisms other than salmon, or its fitness consequences in the wild when it occurs. Here, we hypothesize that larval starvation tolerance is a likely trait subject to domestication selection and experimentally measure and compare this trait in wild and selected-strain oysters.

Natural populations of the eastern oyster *Crassostrea virginica* are only a fraction of historic densities, and are deemed functionally extinct in some regions of the northeastern U.S. [12]. The great ecological and economic value of oysters has prompted aggressive population supplementation (= "restoration") programs in parts of its native range [13,14] and millions of U.S. dollars have been spent to revive this once thriving ecosystem engineer [15]. Oyster restoration approaches often include the hatchery production and planting of spat (juvenile oysters) on shell to supplement wild recruitment and help jumpstart a population on restored habitat. To reduce genetic bottlenecking and other genetic changes in culture, the Nature Conservancy recommends using fresh wild broodstock when producing spat to be deployed for stock enhancement [16]. In practice, however, selectively-bred eastern oyster lines sometimes get used in population supplementation for several reasons. In some regions, "wild" broodstock (i.e. non-feral oysters) are locally sparse and logistically challenging to collect from remnant populations, or available seed oysters from regional production hatcheries only include domesticated oyster lines intended for aquaculture production. Also, in some cases selectively bred traits such as disease resistance are deemed desirable or necessary for successful restoration [11,17,18]. Genetic assignment tests have generated mixed results documenting spat recruitment from hatchery-produced selected strain cohort plantings [19,20]. Trade-offs between aquaculture traits and fitness in the wild have not been evaluated in bivalves to our knowledge.

Concerns with using artificially selected lines in restoration include the degree to which selective breeding inadvertently alters non-target traits, either because they are genetically correlated with the selected trait, or because culture conditions impose selection (domestication selection). Hatchery production of bivalves can also result in increased levels of inbreeding compared to wild populations due in part to the reduced number of breeding parents and

sweepstakes reproduction in which an even smaller number of parents actually contribute to the cohort [14,21–23]. Additionally, hatchery culture imposes a genetic bottleneck relative to wild cohorts and domestication selection further strengthens genetic drift, reducing genetic diversity to a degree that may have a fitness cost under natural conditions [8,10,14]. Evolution of domestication traits is increasingly under study to understand their rate of change and fitness impacts [5–8,24]. Not surprisingly, phenotype and performance data for oysters is mostly restricted to commercially important traits during grow-out from planted spat to adult (market size). Fitness trade-offs across other parts of the life cycle are relatively unexplored, yet larvae are arguably the most sensitive life stage [25,26]. Fitness trade-offs are extremely challenging to study in species with a complex life cycle that includes pelagic larvae [27,28], but it is a reasonable assumption that traits conferring higher relative fitness in culture, at high population density in a homogeneous environment, will not increase mean population fitness in the wild.

For many marine species, including oysters, tremendous fecundity (e.g. millions of eggs per female) and high, non-random early mortality suggests that the larval stage may be under particularly strong selection pressures [26,29,30]. High early mortality (Type-III survivorship) often characterizes these species in culture as well as in the wild [25,26], potentially involving strong selection in culture that could result in directional shifts in the mean value of traits favored by the hatchery environment. In contrast, early mortality due to the expression of a high genetic load may swamp the signal of directional selection on larval traits, or the two mechanisms could interact if the segregation of mildly or strongly deleterious mutations contributes to fitness or growth rate differences among larvae [22,31,32]. Therefore, understanding the selective forces underlying early mortality in the hatchery is particularly important in the context of supportive breeding for restoration, but these mechanisms also are relevant for optimization of selective breeding.

Nutrition is the major driver for growth and development during pelagic life stages of bivalve larvae [33,34]. Natural temporal and spatial variation in food quality and quantity can be extreme [34], leaving free swimming larvae vulnerable to periods of insufficient food supply [35]. During the hatchery culture of bivalves, environmental conditions are controlled to support the best growth and survival at a relatively high larval density, including a simple (low diversity) diet [36]. Because of the high variability of food quality and quantity in the natural environment compared to *ad libitum* feeding under hatchery culture, metabolic processes may be under different selection pressures resulting in domestication selection. In a hatchery-based larval culture experiment in the Pacific oyster, Plough [37] showed that rearing with a 3-species algal diet significantly increased larval growth and survival, and reduced the expression of genetic load (measured as genetic inviability) compared to a single species algal diet, highlighting the importance of food quality specifically and genetic by environment interactions more generally. Therefore, we hypothesize that larvae produced from wild lines would have a greater tolerance to starvation than selectively bred aquaculture lines, which have be reared for many generations with *ad libitum* feeding under hatchery contitions. In order to explore the potential for domestication selection, we experimentally measured and compared this trait in wild and selected-strain oysters. In addition, broodstock from two distinct lineages for the seletectively bred strain were used to reduce the potential for inbreeding to interfere with the phenotypic response.

Oysters produced for the aquaculture industry are often selectively bred for traits that speed up production, such as fast growth, and that improve survival (e.g. disease resistance; [38]). However, it is unclear whether or not other (unintended) traits are evolving due to genetic correlations, adaptation to the artificial environment, or heritable epigenetic changes during hatchery culture. As a first step towards examining the potential effects of domestication

selection in oysters, we performed an experiment comparing the starvation resistance of larvae produced from wild (no prior hatchery exposure) and artificially selected (over multiple generations of hatchery propagation) broodstock oysters. Feeding environment during early development matters a great deal for both aquaculture and population supplementation goals, making starvation resistance an appealing first target among the many traits that could have changed as a result of adaptation to hatchery conditions. Because oyster larvae have a wide variance in growth rates within families [32] and hatchery production often includes the culling of slow growing larvae, we also separated and compared slow and fast early growth larval phenotypes in each line. Separating larvae by early growth rate provided the opportunity to compare stress responses of physiologically distinct portions of each line. We used growth, survival, and respiration (rate of oxygen depletion) as measures of physiological response to a prolonged (10-day) starvation period between lines and among cohort growth-fractions.

## Methods

### Broodstock conditioning and spawning

All work was completed under an institutional permit for collection of oysters under the annotated code of Maryland, Article: Natural Resources, section 3–403, Par.9, 1997 Cumulative Supplement. Wild adult oysters were collected from the Choptank River, Maryland in the Chesapeake Bay and two disease-resistant aquaculture lines (Deby (DBY) and DBY-CROS-breed (XB)) were obtained from the Virginia Institute of Marine Science, Aquaculture Genetics and Breeding Technology Center (ABC). All broodstock were held under chilled (20°C) flow-through Choptank River water at the Horn Point Laboratory Oyster Hatchery in Cambridge, Maryland to promote gametogenesis, but prevent spontaneous spawning. Local salinity was 9–11 ppt during the conditioning period for Choptank River broodstock. Aquaculture (DBY and XB) oysters were partially conditioned at the ABC at a salinity of 14–16 ppt, before being shipped to Horn Point Laboratory where they were held under conditions described above for four weeks prior to spawning. The DBY and XB lines have been bred over multiple generations with hatchery propagation and intensive selection for disease resistance (both MSX and dermo; [38,39]). Broodstock originated in 1998 from Virginia Institute of Marine Science and the Haskin Shellfish Research Laboratory for DBY and XB, respectively, but have since been interbred with Chesapeake Bay oysters and broodstock from Louisiana known to have naturally acquired dermo resistance [38].

On June 26, 2017, oysters were spawned by raising the water temperature from 20°C to 30°C in individual containers with flow-through seawater at a salinity of 9.9 ppt. When temperature did not induce spawning, heat-killed sperm was added to stimulate spawning. As individuals began to release gametes, the water flow was stopped for that individual, time was noted, and sex determined by assessing the released gametes of each individual microscopically. Oysters were allowed to finish spawning in their individual container to collect and isolate gametes for each individual. A total of six pair-cross fertilizations were completed between two females and three males for each strain type (wild and aquaculture) within one hour of the start of spawning to assure quality of gametes. The DBY and XB selection lines are maintained with methods that limit inbreeding [38], but to eliminate any potential for inbreeding effects here, our experimental aquaculture cohort was created with males from the DBY line and females from the XB line (hereafter referred to as AQF1).

Approximately one-hour post fertilization, developing embryos were enumerated microscopically for each pair cross and the number of embryos from each pair were equalized to give each parent pair equal opportunity to contribute to the cohort. For each line, the pair-cross embryos were pooled, then split in half and reared in duplicate 200-L tanks at a density

of 30 larvae mL$^{-1}$ for seven-days using 0.5 μm filtered Choptank River water at 9.7 ± 0.1 ppt and 27°C. Larvae were fed a diet of 50:50 *Isochrysis galbana* and *Chaetoceros calcitrans* beginning at 10,000 cells mL$^{-1}$ on day one and was increased each day by 10,000 cells mL$^{-1}$. Water changes were completed on day three using a 25 μm sieve to assure no larvae were lost and re-stocked at a density of 15 larvae mL$^{-1}$ by random size culling. Water changes were completed every other day thereafter with no culling until day seven. All water was pumped from the Choptank River through a sand filtration system down to 2 μm followed by successive string and cartridge filtration to 0.5 μm.

## Total organic carbon (TOC) analysis

To gain a better understanding of the potential sources of dissolved organic matter (DOM) that may be available as a food resource to larvae during the starvation period, water samples were collected directly from the Choptank River, from the 0.5 μm river water filter system used to fill tanks, from the 0.5 μm filtered water with additional carbon filtration, and from experimental buckets (fed and starved treatments) during a water change (24 hours after filling) 8 days into the starvation period. Water samples were collected in duplicate and filtered through a 0.2 μm glass-fiber filters and shipped to the University of Maryland Center for Environmental Science, Chesapeake Biological Laboratory for total organic Carbon (TOC) and total dissolved Nitrogen (TDN) quantification.

## Starvation challenge

At age 7 days old (July 3, 2017), larvae from each line were separated into fast and slow early growth cohorts by size selecting on an 85 μm sieve. Under normal hatchery rearing, some culling of small larvae (slow growers) is likely to occur inadvertently by age 7 days as a result of increasing sieve size to remove dead shell during water changes. All larvae that were caught on the 85 μm sieve were deemed fast growers and all larvae that went through the sieve, slow growers. This allowed for an approximately equal split in numbers between the fast and slow early-growth fractions for each cohort, although there was overlap in the resulting size distributions (Fig 1). The slow early-growth group includes larvae that would typically be culled during normal hatchery practice.

Larvae were stocked at approximately 15 larvae mL$^{-1}$ in 20 L buckets (~300,000 larvae / bucket) using 0.5 μm filtered Choptank River water at ambient salinity (9.4 ± 0.7 ppt) and temperature (27.6 ± 0.4°C). For each cohort (Wild/Fast, Wild/Slow, AQF1/Fast, AQF1/Slow) two treatment conditions were maintained; starved and fed with four replicate buckets for each cohort/treatment (Fig 2). Fed control buckets were fed according to the Horn Point Oyster Hatchery protocol with an increase in phytoplankton concentration by 10,000 cells mL$^{-1}$ each day and starting at 70,000 cells mL$^{-1}$ on day 1 of the experimental period. A live phytoplankton diet of 50:50 *Isochrysis galbana* and *Chaetoceros calcitrans* was maintained throughout the entire experiment so as to keep conditions constant between fed controls and recovery of the starved treatment. Starting on day 7 post-fertilization, all phytoplankton were withheld from the starved treatment for ten days after which feeding was resumed. The feeding regime re-started as if the oysters were 8 days old (80,000 cells L$^{-1}$) and increased by 10,000 cells L$^{-1}$ each day thereafter. During the experimental period (starvation and recovery), water changes were completed every 1–3 days at which time live survival counts were completed microscopically using volumetric counts and samples were preserved in formalin for length analysis. To obtain concentrated samples and conduct survival counts, larvae were transferred to 500 mL beakers, mixed well and four replicate samples from each beaker were counted to improve precision. Pictures of preserved larvae were taken using a Nikon Eclipse E600 microscope (Nikon[®]

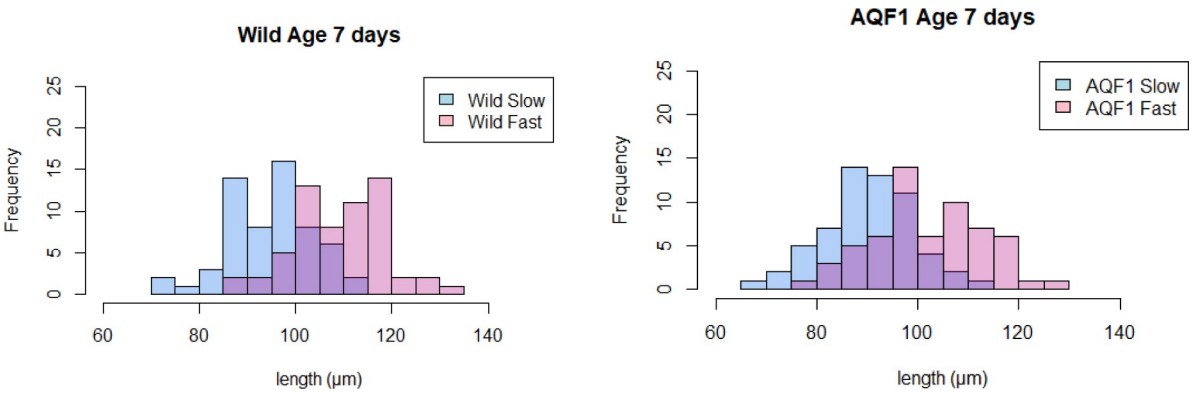

**Fig 1. Length distributions of larvae at the start of the starvation experiment (T = 0; Age = 7 days).** Pink bars represent the fast-growing larvae and blue bars represent slow growing larvae for each line (wild and AQF1). A total of 60 larvae were measured for each cohort (Wild/Fast, Wild/Slow, AQF1/Fast, AQF1/Slow). The purple region indicates the overlap in size between the two groups for each line.

Instruments, Melville, New York, USA) equipped with an AmScope MU800B digital camera (AmScope© Irving, California, USA) and analyzed for length measurements in the AmScope software. Shell lengths of 30 individuals were measured per replicate bucket, except in cases

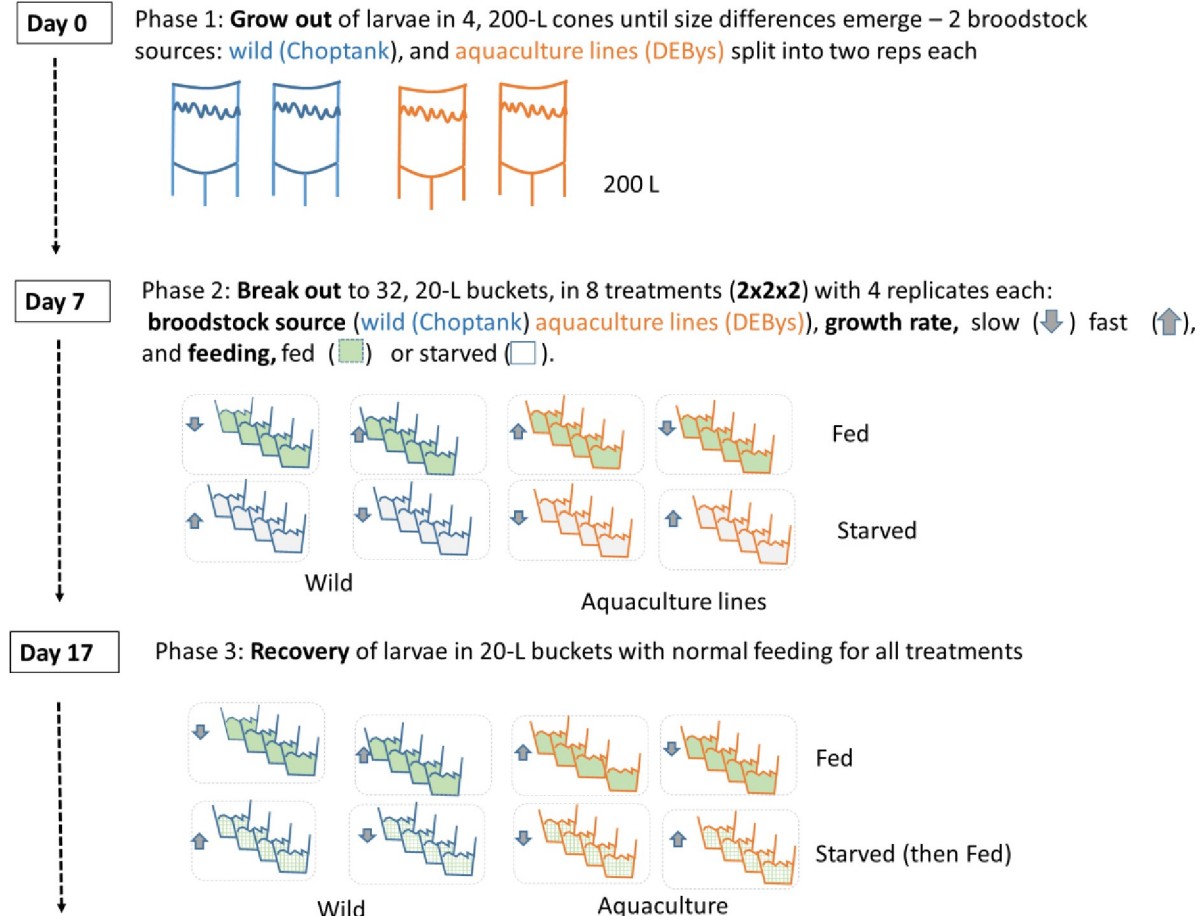

**Fig 2. Schematic of the experimental design.**

where mortality limited the numbers of measurable shells in later sampling time points. When < 10 larvae were found for length measurements, these replicates were removed from growth analysis to avoid sampling error. Growth rates were calculated during the period of linear growth for each treatment and standardized by the number of days for each growth period [growth rate = $(L_{t=2} - L_{t=1}) / (\# \text{ of days})$]. For fed controls, the growth rate was calculated from age 10–17 days and growth rate for the starved treatment was calculated during the recovery period, once growth resumed (age 22–29 days). Due to differences in development and time to settlement, growth rates could not be compared over the same age period between treatments. Changes in tissue coloration and locomotion were noted during live survival counts, but were not quantitatively measured.

## Respiration

Respiration rates of larvae were measured using FireStingO$_2$ fiber-optic oxygen meter (Pyro Science Co., Aachen, Germany) in 5mL closed respirometry chambers. Measurements were made for each of three replicate buckets per cohort/treatment on T = 10, 15 and 18 days after treatment onset (ages 17, 22, 25 days) by transferring 1,000 larvae (volumetric counts) to each vial (200 larvae mL$^{-1}$). Filtered seawater controls were run simultaneously to account for background respiration and vials were kept in the dark to inhibit photosynthesis of any autotrophic organisms present in the culture media (e.g. phytoplankton in the fed treatments). Oxygen depletion was monitored continuously with measurements recorded every minute. Respiration rates are reported as the slope of the linear regression, corrected for background bacterial respiration in the controls, giving a rate of oxygen depletion during the period of linear decline and over a period of one hour, starting approximately 20 minutes after deployment to the chambers to limit noise just after handling. The measurement period was limited to include only measurements before oxygen depletion would lead to disruptions in respiration rate. Due to high mortality in the starved treatment for the AQF1 line (both fast and slow cohorts), replication was lacking and therefore respiration rates were not measured.

## Statistical analysis

A one-way ANOVA was used to test for significant differences in the water chemistry (TOC and TDN) among water sources. Survival and growth were analyzed by two-way ANOVA with length or survival being the response variable, cohort (Wild/Slow, Wild/Fast, AQF1/Slow, AQF1/Fast) and day being independent factors to test for differences between lines and growth cohorts over time. Feeding treatment (starved and fed) was also included as an independent factor for analysis of the first seven days of the experiment. However, due to differences in developmental stage, feeding treatments were also analyzed independently to examine differences among line and growth cohorts. Survival data were arcsine square root transformed to meet the assumptions of normality. Respiration rate was analyzed by two-way ANOVA using cohort, feeding treatment, and day as independent factors. When differences were detected, all analyses were followed by a Tukey's HSD post hoc analysis. Statistical analysis was completed in RStudio version 3.5.2 and significance reported when $p \leq 0.05$. Results are presented as means ± standard error.

## Results

### Spawning and initial cohort attributes

Fertilization success was ≥ 95% for all pair crosses. The number of fertilized embryos produced from each pair ranged from $1.4 \times 10^6$–$9.0 \times 10^6$ and averaged $2.0 \times 10^6$ for AQF1 pairs

and $5.4 \times 10^6$ for the wild pairs. A standardized count of $1.4 \times 10^6$ embryos per pair cross were pooled within each line to give each pair equal chance to contribute.

At the start of the starvation treatment (age 7 days old) fast growing larvae in the AQF1 and wild lines averaged $103.6 \pm 0.6$ μm and $108.8 \pm 1.7$ μm shell length, and the slow growing larvae were $89.4 \pm 1.6$ μm and $94.23 \pm 1.8$ μm, respectively (Fig 1). The coefficient of variation in size was similar across cohorts (ranging from 8–11% among cohorts). Wild/Fast larvae were significantly larger than Wild/Slow and AQF1/Slow (One-way ANOVA $F_{3,4} = 14.97$; $p < 0.05 = 0.012$), however AQF1/Fast had a larger overlap in size with AQF1/Slow and were not statistically different ($p = 0.058$) from each other. Larvae from duplicate culture tanks were pooled before counts, and at that time survival (age 3–7 days) was 60% and 67% in AQF1 and wild lines, respectively.

## Total organic carbon and dissolved nitrogen

The amount of TOC in the hatchery water was lower than that in the river water in this experiment, however large amounts of carbon remained in the hatchery filtered water (CFS) for potential assimilation by larvae (Fig 3A). The addition of the carbon filter significantly reduced the amount of TOC present in the water ($F_{6, 11} = 331.11$; $p < 0.001$; Post Hoc Tukey's HSD $p < 0.001$; Fig 3A). The carbon filter addition contributed to the removal of TDN from the river water, although the change was subtle. TDN was significantly elevated in the fed treatments compared to starved treatments and the initial concentrations in both the River water and filtered hatchery water ($F_{6, 11} = 14.908$; $p < 0.001$; Post Hoc Tukey's HSD $p < 0.001$; Fig 3B).

## Survival

After only 3 days of starvation (T = 3, age = 10 days), clear distinctions in gut coloration were seen with dark full guts in the fed controls and only light coloration in the starved larvae (S2

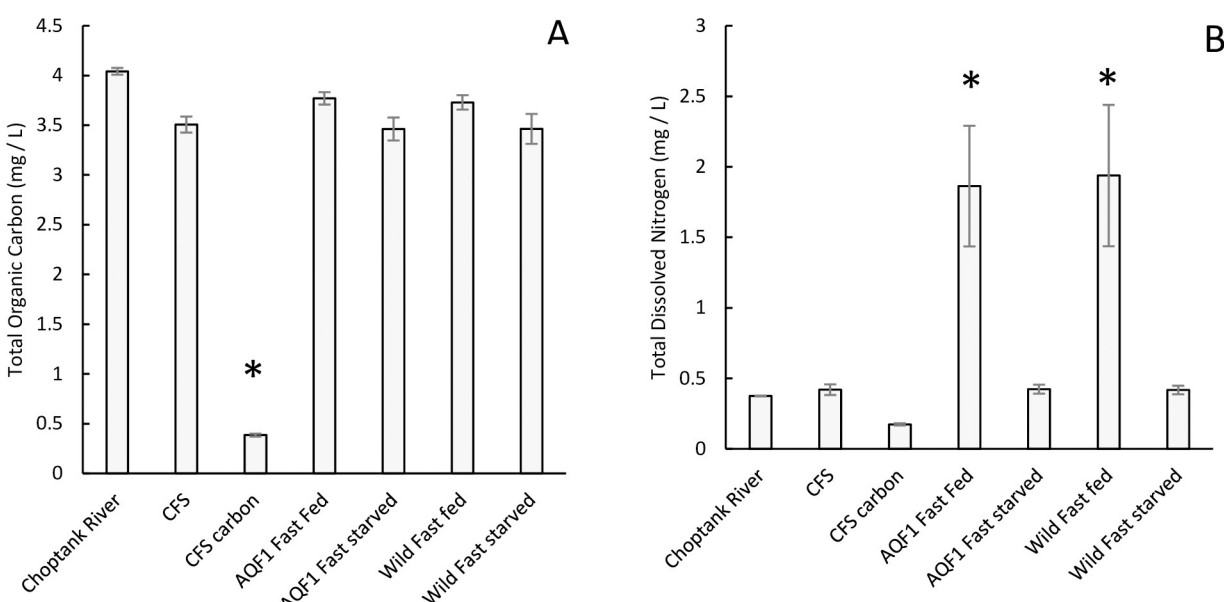

**Fig 3.** Total organic carbon (A) and total dissolved nitrogen (B) in water collected directly from the Choptank River, the Cartridge Filtration System (CFS) used for experiments, and water from the filtration system with an additional carbon filter (CFS carbon). Additionally, water samples were collected from fed and starved treatments during water changes for comparison. Samples from the buckets were taken after approximately 24 hours of larval culture. Asterisks indicated water samples that are significantly different.

Fig). Starved larvae were still actively swimming and casual observations indicated comparable activity to that of the fed controls. After 7 days of starvation (T = 7, age = 14 days), swimming activity was minimal in the starved treatment and guts showed little coloration (S3 Fig). Many larvae in the fed controls developed eye spots by age 14 days, a predictor of settlement competency. By the end of the starvation period (T = 10 days; Age = 17 days) larvae in the starved AQF1 line primarily consisted of dead shell with very few live larvae found in samples for measurements (S4 Fig) and developmental differences had increased between the fed control and the remaining live larvae in the starved treatment. Starved larvae were stunted in the early stages of developing the umbo, while fed larvae had fully developed umbos and many had advanced to develop eye spots. Additionally, settlement was observed on the sides of the fed treatment tanks beginning at age 15 days. During the first seven days of starvation, survival rates showed similar patterns between treatments, with a significant cohort effect ($F_{3, 210}$ = 20.7816; p< 0.000001) and day effect ($F_{6, 210}$ = 18.0786; p< 0.000001), but no significant difference between treatments. The cohort effect was largely driven by the significantly lower early survival rates for Wild/Slow larvae in both treatments (Tukey's HSD; p < 0.05; Fig 4), while survival rates in all other cohorts were similar.

In the fed controls, survival was significantly affected by day ($F_{8, 82}$ = 7.2417; p < 0.05) and cohort ($F_{3, 82}$ = 9.326; p < 0.05), but there was no interaction between them. The Wild/Slow cohort had significantly lower survival than all other cohorts (Tukey's HSD p < 0.01) throughout the experimental period (Fig 4A). By age 14 days, larvae in all fed cohorts showed signs of competency (eyespots and extension of the foot in search of substrate) and setting was observed on the sides of the buckets in the days following this observation. Therefore, survival beyond day 14 could not be used to accurately compare cohorts as reduction in larval numbers was in part due to settlement.

Survival in the starved treatment showed a significant interaction between day and cohort ($F_{33, 133}$ = 2.073; p < 0.005). High early mortality was observed in the first three days before leveling off from age 11–15 days. A second drop in survival was observed toward the end of the starvation period (age 15–17 days), especially for the AQF1 line. By the end of the starvation period (age 17 days) no statistical difference was observed between the Wild/Slow and Wild/Fast cohorts with 19 ± 5% and 23 ± 11% survival, respectively (Fig 4B). Both AQF1 cohorts showed a large drop in survival from age 15–17 days with AQF1/Slow and AQF1/Fast cohorts finishing the 10-day starvation period with only 11 ± 3% and 4 ± 3% survival, respectively (Fig 4B). Mortality continued even after food was reintroduced with < 10% survival after 4 days of recovery (Age = 22 days) for all starved cohorts (Fig 3B inset). Graphical trends suggest that the higher survival observed for wild cohorts at the end of starvation was maintained during the recovery period. However, AQF1 cohorts were each pooled by age 22 days, therefore statistical comparison between lines was not possible due to a lack of replication. Although overall survival was low in the starved treatments, those that survived were successful in completing metamorphosis and eye spot development was observed as early as age 22 days, just four days into the recovery period.

## Growth

For larvae in the fed controls, shell length was significantly affected by cohort ($F_{3, 93}$ = 16.497; p < 0.001) and day ($F_{7, 93}$ = 250.557; p < 0.001) but not their interaction. The significance of cohort as an effect was largely driven by the significantly larger size observed in the Wild/Fast larvae compared to all other cohorts at ages 14–19 days (Tukey's HSD p < 0.001; Fig 5). When comparing growth rates during the period of linear growth (age 10–17 day) in the fed control, there was a significant difference between lines (Wild and AQF1) ($F_{1, 17}$ = 26.4570; $p$ = 0.001).

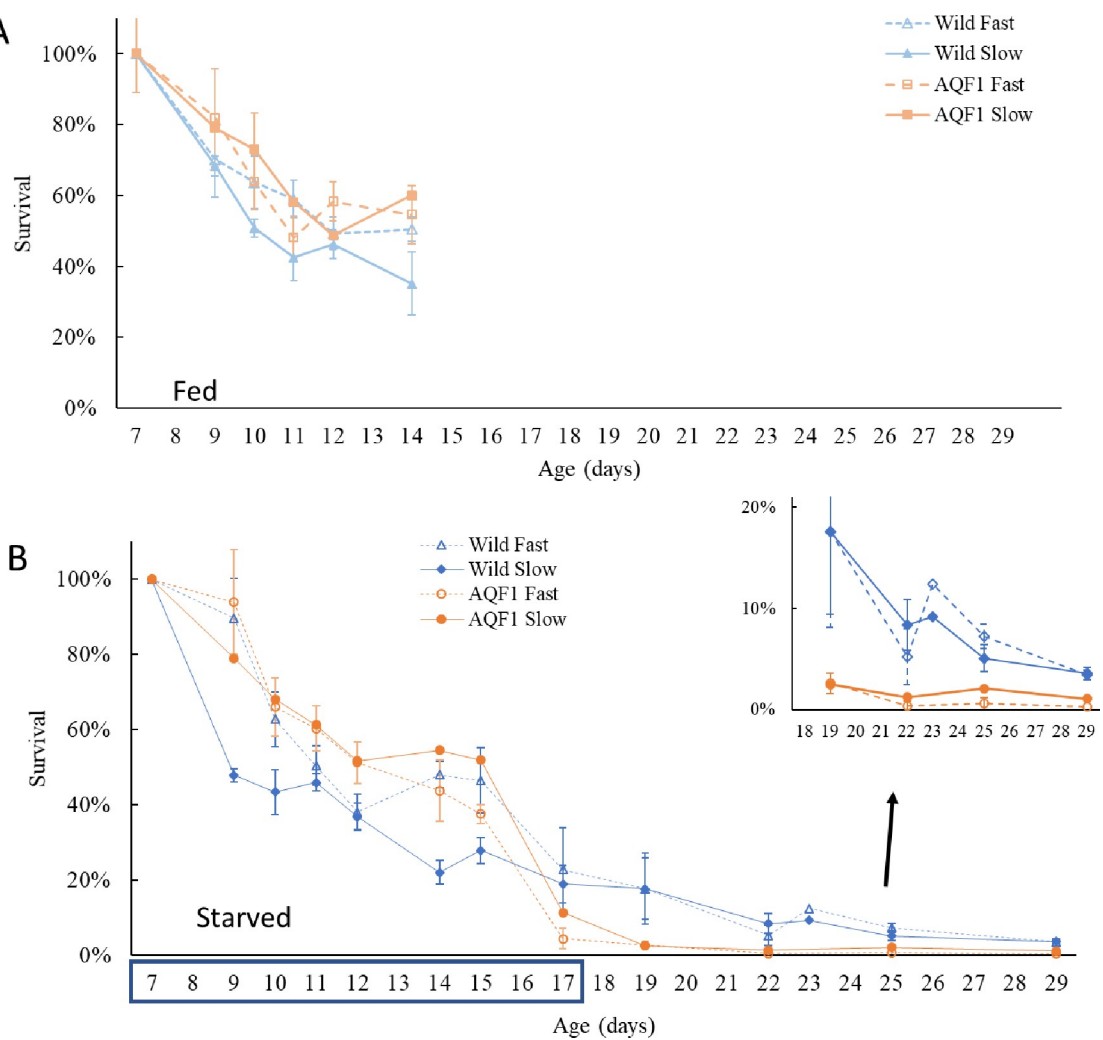

**Fig 4. Survival over time for each cohort.** The starvation period lasted from age 7–17 days (indicated by the blue box on the x-axis) at which time (T = 10 days) food was reintroduced and the recovery period began. Fed controls (top) began setting to the sides of the buckets by 15 days old (T = 8 days) and thus the plot is truncated. Inset shows closer detail of the survival observed for starved larvae at the end of the experimental period (Age 19–29 days). By age 22 days, replicates for AQF1/Slow and AQF1/Fast were each pooled due to low survival within each replicate. Error bars represent standard error.

Wild/Fast larvae had significantly greater growth rates than both AQF1 early growth cohorts (Tukey's HSD p < 0.05), but were similar to Wild/Slow larval growth rates (Table 1). The decrease in mean shell length for Wild/Fast after age 17 days and for all other cohorts after age 22 days, is reflective of high settlement rates of larger larvae which are no longer measured as part of the larval pool. By the end of the experimental period mean shell length for the remaining larvae were similar across cohorts.

Shell length differed significantly between the fed controls and starved treatments with significant treatment by day ($F_{1, 71}$ = 254.39; p = $2.2 \times 10^{-16}$) and cohort by day ($F_{12, 71}$ = 2.62; p = 0.006) interactions during the first seven days of treatment (before settlement started in fed control; Fig 5). All larvae in the fed treatment continued to show significant growth over the first seven days, while growth in the starved treatment was stunted for all cohorts (Fig 5). Within the starved treatments, a significant cohort by day interaction ($F_{21, 83}$ = 2.32; p = 0.004) also was observed across the duration of the experimental period. Shell length showed no

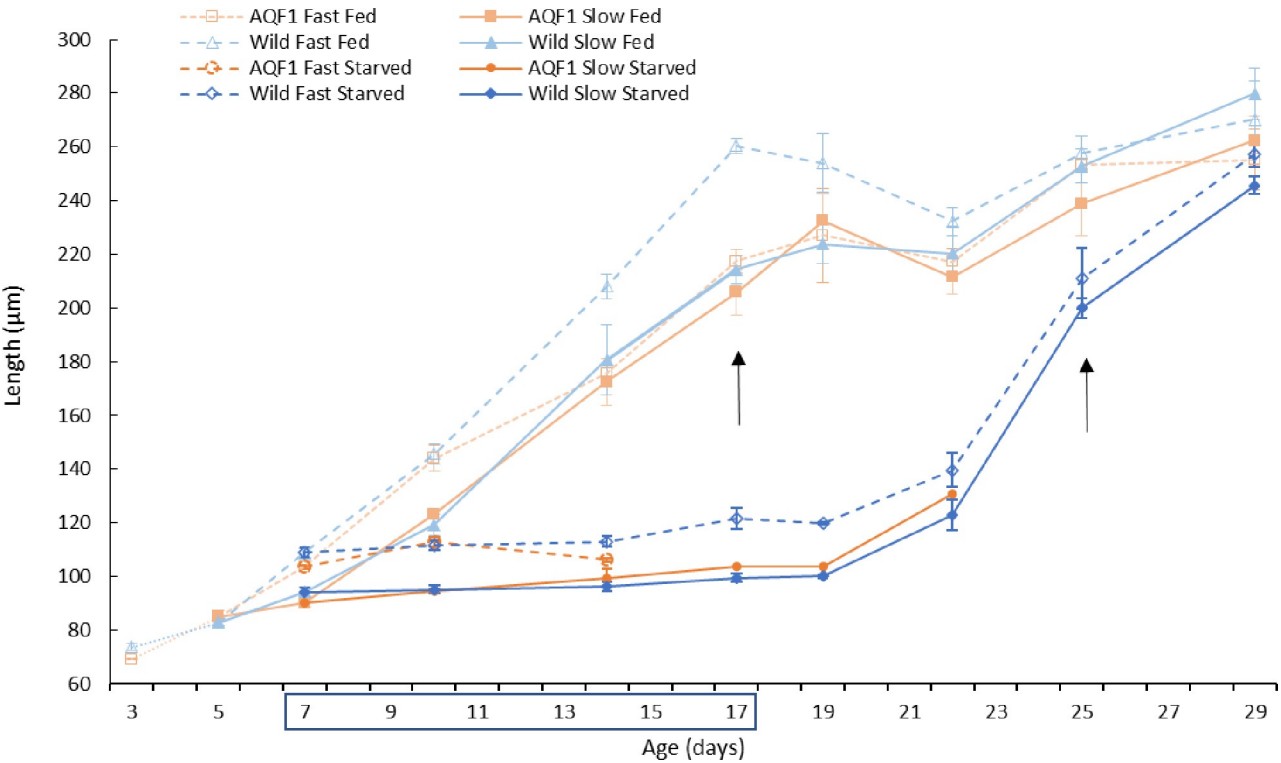

**Fig 5. Mean shell length over time for each line by cohort grouping in fed controls and starved treatment.** Culture from days 3–7 post fertilization occurred in 200-L larval tanks (prior to starvation) and size-based experiments in 20-L tanks. Shell lengths for fed controls after age 14 days and starved treatments after age 22 days should be interpreted with caution because settlement of large individuals out of the larval pool may have biased the size distribution. Arrows indicate days in which settlement was clearly observed in each treatment. Age 7–17 days represent the period of starvation, indicated by the blue box on the x-axis. Error bars represent standard error. Due to low survival in the AQF1 cohorts, length measurements were not possible after age 14 and 22 days for fast and slow growers, respectively. Additionally, from age 17 to 22 days in the AQF1/Slow cohort only one replicate of length measurements was possible.

significant increase once food was removed, indicating shell growth had stopped and fast early-growth cohorts remained larger than slow early-growth cohorts. However, we did observe a decrease in mean shell length in AQF1/Fast during the age 10–14 days interval. This size decrease was associated with high mortality in the starved AQF1/Fast cohort, but interestingly, it occurred in the same interval when the fed AQF1/Fast cohort had a growth rate deceleration (Fig 5). Even after food was reintroduced to the starvation treatment, growth lagged for at least two days (age 17–19 days) in all cohorts before starting to increase (age 19–22 days), ultimately achieving growth rates similar to the fed treatment between day 22 and 29 (during which settlement also started; Fig 5). Once growth resumed during the recovery period, Wild/Fast and Wild/Slow had no significant difference in shell length (Fig 5). For the AQF1 line, high mortality in the starved treatments prevented growth comparisons after day

**Table 1. Growth rate (μm/day ± SE) during the linear growth phase for fed (age 10–17 days) and starved-recovering (age 22–29 days) larvae.** AQF1 starved cohorts were not included in the statistical analysis due to high mortality and a lack of replication (one pooled replicate per cohort) during the recovery period. Superscript letters indicate significant differences between cohorts within each treatment and asterisks (*) indicate significance between treatments within each cohort (Tukey's HSD $p < 0.05$).

|  | AQF1/Slow | AQF1/Fast | Wild/Slow * | Wild/Fast |
|---|---|---|---|---|
| Starved 22–29 days | 15.49 | 8.7 | 17.55 ± 1.1 | 16.95 ± 0.9 |
| Fed 10–17 days | 12.04 ± 1.1 [B] | 10.46 ± 0.6 [B] | 13.63 ± 0.8 [AB] | 16.23 ±0.4[A] |

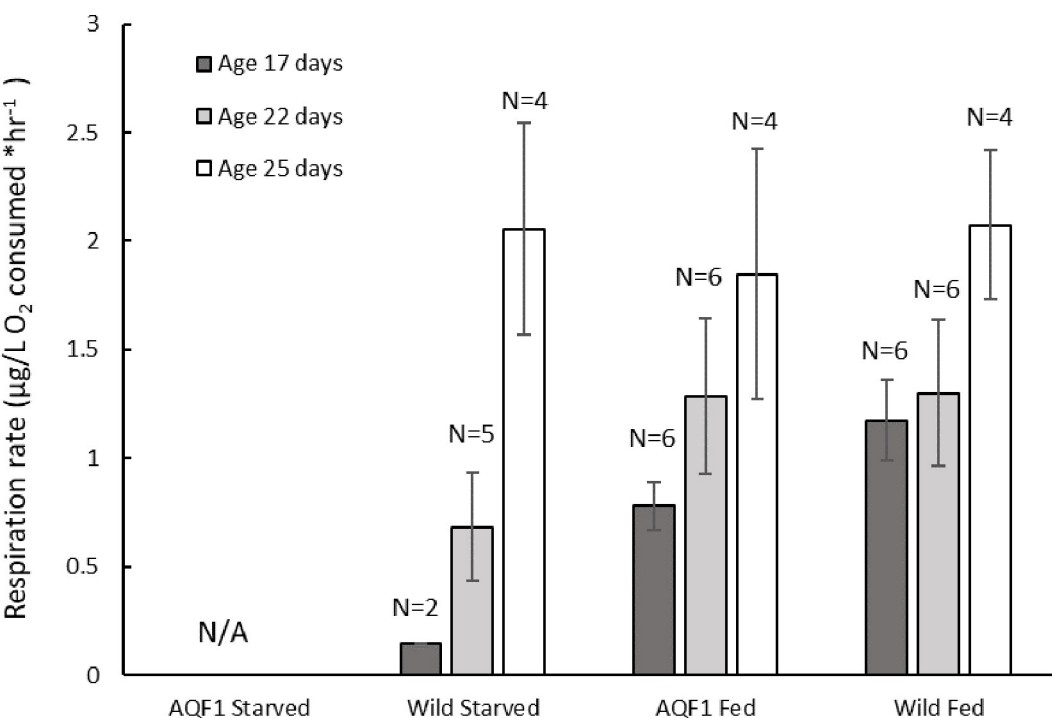

**Fig 6. Respiration rates of all cohorts measured at the end of the starvation exposure (T = 10; Age 17 days), and during recovery at 5- and 8-days post reintroduction of food (Age 22 and 25 days, respectively).** The number of replicates per time point for each cohort is indicated above each bar. Error bars represent standard error.

14 when less than 10 larvae were found for measurement in all replicates except for one replicate in the AQF1/Slow cohort. When comparing the growth rate of the wild larvae across early growth cohorts and treatments, a significant treatment effect ($F_{1,8} = 7.7860$; $p = 0.0235$) was observed, which was driven by the significantly greater growth rate in the Wild/Slow starved treatment during the recovery period (age 22–29 days) compared to the linear growth period for the Wild/Slow fed treatment (Tukey's HSD; $p = 0.0317$). In contrast, no significant difference was observed between treatments for the Wild/Fast cohort (Table 1). A one-way ANOVA with wild larvae endpoint data (age 29 days when both groups were losing larger individuals to settlement) showed no significant difference in shell length between treatments although the trend was for starved treatment larvae to have smaller size (251.0 ± 3.4μm) on average compared to fed controls (267.0 ± 6.2μm).

## Respiration

Due to low replication, Fast and Slow cohorts for each line were pooled for comparison of respiration rates between lines. Respiration rate was significantly affected by day ($F_{1, 38} = 13.8952$; $p < 0.001$) and increased with age for both fed control and post-starved larvae (Fig 6). Larvae at age 25 days had significantly higher respiration rates than both 17 and 22 days (Tukey's HSD; $p < 0.05$). There was not a significant effect of treatment, but this may be due in part to low replication (N = 2) in the starved treatment for measurement at age 17 days. Respiration rates at the end of the 10-day starvation period were low compared to the fed controls (Fig 6), but within 5 days of the reintroduction of food the Wild/Starved larvae had rates similar to that observed in the fed controls.

## Discussion

The distribution of genomic and phenotypic changes during domestication are a fundamental question related to breeding practice in aquatic species [5–7], with potential fitness impacts on wild populations if there is interbreeding or deliberate population supplementation with domesticated lines. As a first step to investigating the potential for domestication selection in the eastern oyster, we compared starvation tolerance of larvae produced from wild and selectively bred broodstock. Assuming that selection for this trait is strong in the wild and nonexistent under culture, we predicted that starvation tolerance would be greater in the larval progeny of wild oysters relative to larvae from closed selection lines (multiple generations of larval culture with *ad libitum* feeding). This prediction was partially informed by the existence of considerable molecular transporter machinery that allows for larval body maintenance via DOM absorption in the absence of algal food [40]. If this transporter machinery is energetically costly, starvation tolerance could weaken in closed selection lines as either a response to release from selection, or to selection for faster growth. For example, selection for faster growth under *ad libitum* feeding could have reduced expression of these proteins, lowering starvation tolerance. Alternatively, selection for faster larval growth could have generated changes in other metabolic traits with DOC transporter functions maintaining their fitness value due to pleiotropy even under *ad libitum* feeding. These scenarios suggest that selected strain larvae should grow faster than wild larvae under *ad libitum* algal feeding, all else being equal and assuming no inbreeding effects. If a fast-growth selected strain does not show faster larval growth on a live phytoplankton diet, this might suggest weak selection on this life history stage in the history of the selected strain, and in this case unequal starvation tolerance would imply correlations with other traits under stronger selection.

Control larvae fed *ad libitum* phytoplankton showed overall similar mortality and growth rates between lines. Aquaculture larvae in this experiment were F1 hybrids between two disease resistant lines, so no inbreeding depression was expected and apparently no heterosis (hybrid vigor) was induced. This parity contradicts predictions that selectively bred aquaculture lines have greater larval growth rates compared to wild larvae. Despite this parity between lines early in the experiment, by the end of the 10-day starvation period (age 15–17 days), wild larvae showed significantly greater survival than the AQF1 line, consistent with our domestication effect prediction. Starvation consisted of withholding the preferred food (phytoplankton), but micronutrients in the form of dissolved organic matter (DOM) remained available to the larvae in both the fed and starved treatments, and presumably were utilized to maintain viability. If AQF1 lines lost some of the wild capacity to utilize DOM to maintain body condition under food limitation, it suggests that there may be an energetic cost to the molecular transport machinery for providing DOM uptake, and release from selection due to unlimited provisioning, or selection for fast larval growth could lead to selection against those transport mechanisms if they have few other functions. Testing these more specific mechanistic hypotheses, and their impact on fitness, will require further study. Given that this is the first test attempting to understand mechanisms that could be linked to domestication in eastern oysters, that we are aware of, we discuss caveats and implications for these findings in the context of related literature.

### DOM as a source of nutrients during starvation

This study focused on a larval trait predicted to affect fitness in the wild and be subject to inadvertent selection in hatchery culture (or release from selection maintaining tolerance). Specifically, we compared starvation tolerance in larvae produced from wild parents and larvae produced from selectively-bred parents. The selectively bred AQF1 line experienced nearly

complete mortality during the last three days of starvation, while both fast and slow wild cohorts experienced better survival. Analysis of dissolved organic carbon suggests that although particulate food (phytoplankton) was withheld, micronutrients (e.g. DOM) were present during starvation. For adult zebra mussels feeding on natural seston, DOC contributes up to 50% of the carbon demand in adults [41] and it is well documented that bivalve larvae are capable of utilizing natural sources of DOM [40,42–44]. Therefore, using DOC as a proxy for DOM, it is plausible that DOM remaining in the starved treatment water may have provided the energy needed to fuel basal metabolism requirements during the "starvation" period. While DOM alone is not expected to sustain growth and development, it might allow for somatic maintenance under food limitation [44] and is expected to represent a large potential energy source for developing larvae under natural conditions [40]. We therefore hypothesize that the ability to absorb, transport, and assimilate micronutrients may be an important source of differentiation between the lines tested and represent promising phenotypes to explore as possible mechanistic changes associated with domestication selection. It is plausible that aquaculture lines can adapt to the hatchery environment, in which food is provided in excess, thereby limiting their ability to utilize DOM during prolonged starvation events as a result of many generations of hatchery propagation. However, this requires further and more detailed testing to understand the role of DOM under food limitation.

## Survival during prolonged starvation

The most dramatic mortality was observed between days 8 and 10 of the starvation (age 15–17 days) in the AQF1 line suggesting that a critical physiological threshold was reached. Blaxter and Hempel [45] described a point of no return, in which the duration of starvation induces an irreversible physiological toll, resulting in death even if a proper food source is restored. This is consistent with the continued mortality we observed during the recovery period, but clearly there was individual variation in this threshold because some larvae did recover. Survival continued to decline through age 22 days (5 days into the recovery), during which growth also remained stunted, suggesting that larvae had not yet rebounded physiologically. When starvation begins at the time of hatch, His and Seaman [46] suggest that the point of no return for *C. gigas* larvae occurs when maternal reserves are depleted (6–8 days post-fertilization), but Moran and Manahan [47] observed no significant change in mortality rate during a starvation period up to 14 days post fertilization. In our study, both wild and AQF1 lines had lower starvation tolerance than that reported by Moran and Manahan [47], perhaps due to the different age of onset for starvation. When starvation occurs from onset of hatch, larvae can utilize lipid reserves provided by the egg and reduce metabolic rates allowing for long term survival (up to 14 days) [34,47]. However, when food is removed after egg lipid reserves are depleted (6–8 days post hatch), as in our study, mortality can be high even under short periods of starvation (e.g. 4 days) [34]. All previous oyster work on these questions has used *C. gigas*, so there may also be species differences explaining variation in findings.

   Larval survival and successful recruitment of subsequent generations following plantings of hatchery produced spat or adult oysters is an essentential component to long term restoration success. If planted oysters from selective breeding programs produce larvae that cannot survive the gauntlet of stressors in the estaurine environment, then long-term restoration success is stymied. While we acknowledge that a 10-day period of no phytoplankton availability may be unlikely under natural conditions, patchiness in larval food quantity [35] and nutritional quality [47] are expected. Under these conditions, an ability to withstand and recover from periods with limited exogenous energy sources is a critical fitness trait. The lower tolerance to starvation in the selected strain studied here is consistent with this trait being a costly adaptation

weakened as a result of selection for fast growth or, more likely, a trait correlated with commercial traits under direct selection. Further comparisons with other eastern oyster selected lines are needed to determine if this larval tolerance difference is a general result of domestication or specific to the tested lines.

## Physiological recovery

We observed a two to five-day delay in shell growth once food was reintroduced to the starved treatments. This is in contrast to Moran and Manahan [47] who saw an immediate resumption of physiological rates, including growth, at the onset of delayed feeding of *C. gigas* larvae. The delay in growth we observed during the recovery period may be due in part to a delay in recovering normal feeding behavior [46] and the rebuilding of lost tissue mass and energetic reserves before energy is used for shell growth [48]. We observed reduced respiration rates at the end of the 10-day starvation period, but within 5 days of the recovery period respiration rate was similar to that observed in the fed control groups. Shell growth during the first 5 days of recovery was low, suggesting that although there was an observed recovery in respiration, there was likely a priority given to somatic tissue growth prior to the production of new shell. While not quantified, larvae from both lines had a visible loss of tissue mass and an inhibition of locomotion during starvation, suggesting that any active metabolism was only to maintain homeostasis during a depressed metabolic state. However, even after 10-days without food, respiration rates in both wild fast and slow cohorts were still measurable, suggesting that they may be utilizing some exogenous energy source (e.g. DOM; [47]). After eight days of recovery, growth (as measured by mean shell length) in the starved treatments was similar to that observed in the fed controls and starved larvae reached final shell lengths similar to that observed in the fed controls during peak settlement showing a full recovery for those that survived the starvation period.

## Variation in growth cohorts between lines

Starvation tolerance is a complex trait, so we expected there might be interactions with growth rate. Given the development of high size variance early in each line (typical of eastern oysters), we separated each line into fast and slow early-growth cohorts to test for relationships with starvation tolerance. The only study that previously examined oyster larval growth variance [32] did not separate and follow individual size fractions. However, using size separation, Bitter et al. [29] showed fast growing larvae to have different patterns of selection than slower growing larval groups when experiencing environmental extremes. Even though distinct growth fractions might be better separated later in larval culture, for the purposes of this experiment, the separation was quite early, at day 7 post-fertilization. If the fractions represent differences that only affect early larval growth then their subsequent growth trajectories were expected to be parallel and an interaction with starvation tolerance was less likely. Alternatively, if early growth differences marked a persistent phenotypic difference then distinct growth rates were expected for slow and fast fractions of each line. For the wild larvae in the fed treatment, the initial size separation between fast and slow growth cohorts was maintained throughout the experimental period, with no significant difference in growth rate observed. In the fed AQF1 line, the initial size distinction was maintained up to age 10 days (3 days after size separation), but depressed growth of AQF1/Fast led to size convergence with AQF1/Slow by age 14 days. The reasons for changes in growth rate in AQF1/Fast (seen in both fed controls and the starved treatment) are not known, but they were apparently specific to a particular developmental stage and do not compromise the main finding of line growth rate parity.

The size difference between fast and slow cohorts was maintained in the starved treatments throughout the starvation period. Interestingly, once food was reintroduced, the significant difference in length between growth cohorts disappeared for wild larvae. This is in part due to a slightly faster growth recovery in the Wild/Slow cohort during age 19 to 22 days, during which CV for length increased for both wild cohorts suggesting that some larvae rebounded more rapidly than others (S1 Fig). The convergence of growth trajectories between fast and slow Wild larvae during recovery is suggestive of compensatory growth in the Wild/Slow cohort, however size specific mortality and/or settlement cannot be ruled out as interval mortality was high during this time (30% and 48% for fast and slow, respectively) and settlement was apparent in both cohorts by age 22 days.

## Impacts of line history on the experimental design

There are a number of additional factors associated with the design and setup of this experiment that may have influenced the outcome and are important to consider. Inbreeding depression in the AQF1 line is an unlikely explanation for their relatively low survival late in the starvation period because two largely independent selected lines were crossed to produce the AQF1 larvae for this experiment. In fact, heterosis effects were a potential outcome of this mating strategy [49], but were not apparent. Differential maternal effects can also impact larval survival and response to stress given the importance of egg lipids to early larval development [50]. However, all broodstock were held under local ambient conditions for four weeks prior to spawning in order to reduce the impact of different environmental conditions on gamete quality. Also, the experimental treatment was delayed until day seven post-fertilization to reduce the potential for differences in maternal energy reserves to impact larval tolerance to starvation [46]. Broodstock source salinity is also a potential confounding factor in this experiment. The aquaculture lines were partially conditioned at a higher salinity (14–16 ppt) prior to arriving at Horn Point where all broodstock were held for four weeks at the experimental salinity (9.5 ppt) prior to spawning. This did not seem to put the AQF1 line at a developmental disadvantage because growth was similar among the fed controls and survival of AQF1 larvae was similar or better than the Wild line. Also, the aquaculture lines used in this experiment have previously performed well under low salinity conditions (9–15 ppt) [39]. Salinity during broodstock gonad conditioning is known to have transgenerational plastic effects on salinity tolerance in the larval offspring [51,52], so it is possible that the reduced salinity at the end of the conditioning period could diminish larval resistance to stress in the AQF1 line. Lastly, we note that because of extensive restoration in the Chesapeake Bay that includes seeding with hatchery produced spat [13,14], it is conceivable that our wild broodstock oysters could have some ancestry from hatchery-produced restoration oysters. However, oysters produced in this region would have been sourced from the Horn Point Oyster Hatchery where wild broodstock are used for supportive breeding. Therefore, broodstock collections used here would at most only have one prior generation of hatchery propagation, compared to many generations in the closed aquaculture lines.

## Conclusions

For oysters, lines artificially selected for aquaculture are serving the farming industry, where lifetime fitness is not a focus, but rather the emphasis is on farm to table production rates in which yield is largely a function of growth rate and survivorship. However, in the context of using hatcheries for stock enhancement, planted oysters must not only grow to maturity, but also successfully reproduce and generate offspring with robust abilities to withstand the many stressors encountered during the pelagic life stage. At this early stage of selective breeding in

eastern oysters, the unknown phenotypic impacts from domestication selection may be large or small, depending on the trait. In this initial experiment, we have shown a slightly lower tolerance to prolonged starvation in the AQF1 line intentionally selected for disease resistance over multiple generations of hatchery propagation. Replication of this experiment will be necessary using more biological replicates, different aquaculture lines, and additional wild oysters to strengthen the inference that reduced larval starvation tolerance is linked to domestication selection generally. Based on these initial results, we suspect bioenergetic processes related to micronutrient uptake and utilization may be promising candidate traits for investigating mechanistic changes as a result of domestication selection, or are genetically correlated with other traits under direct selection. The great successes of selective breeding to produce shellfish with improved aquaculture yields is likely to result in an increased reliance on selectively bred lines for oyster farming. With further oyster domestication expected, our work is of importance to understand inadvertent trait evolution as well as the potential impacts domesticated oysters can have on natural populations.

## Supporting information

**S1 Fig. Box plots showing length distributions over time for each treatment cohort.** After age 14 days, the starved aquaculture growth fractions have very few measurements and means may be skewed due to the high mortality rates.
(DOCX)

**S2 Fig. Micrographs of 10-day old larvae in each treatment after three days of starvation.**
(DOCX)

**S3 Fig. Micrographs of 14-day old larvae in each treatment after seven days of starvation.**
(DOCX)

**S4 Fig. Micrographs of 17-day old larvae in each treatment after ten days of starvation.**
(DOCX)

## Acknowledgments

We would like to thank the Horn Point Laboratory Oyster Hatchery for technical support during this work and conditioning of broodstock. Thanks also go out to Katherine Hornick for technical support during sampling. We would also like to thank two anonymous reviewers for careful review and comments which helped improve the manuscript.

## Author Contributions

**Conceptualization:** Katherine McFarland, Louis V. Plough, Matthew P. Hare.

**Data curation:** Katherine McFarland.

**Formal analysis:** Katherine McFarland, Michelle Nguyen, Matthew P. Hare.

**Funding acquisition:** Louis V. Plough, Matthew P. Hare.

**Investigation:** Katherine McFarland, Louis V. Plough, Michelle Nguyen.

**Methodology:** Katherine McFarland, Louis V. Plough, Michelle Nguyen, Matthew P. Hare.

**Project administration:** Katherine McFarland.

**Resources:** Louis V. Plough, Matthew P. Hare.

**Supervision:** Katherine McFarland, Louis V. Plough, Matthew P. Hare.

**Visualization:** Katherine McFarland, Michelle Nguyen.

**Writing – original draft:** Katherine McFarland, Louis V. Plough, Michelle Nguyen, Matthew P. Hare.

**Writing – review & editing:** Katherine McFarland, Louis V. Plough, Matthew P. Hare.

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
