## [Decision Letter · Decision Letter 0]

13 Apr 2020

PONE-D-20-05015

Are bivalves susceptible to domestication selection? Using starvation tolerance to test for potential trait changes in eastern oyster larvae

PLOS ONE

Dear Dr. McFarland,

Thank you for submitting your manuscript to PLOS ONE. Your manuscript was evaluated by two experts. Based upon their constructive and thoughtful comments, I am recommending publication after minor revision. I am confident that the reviewers' suggestions and concerns can be readily addressed in a revised version. Please, notice the attached document with  edits from one of the reviewers.

We would appreciate receiving your revised manuscript by May 28 2020 11:59PM. To enhance the reproducibility of your results, we recommend that if applicable you deposit your laboratory protocols in protocols.io, where a protocol can be assigned its own identifier (DOI) such that it can be cited independently in the future. For instructions see: http://journals.plos.org/plosone/s/submission-guidelines#loc-laboratory-protocols

We look forward to receiving your revised manuscript.

Kind regards,

Hans G. Dam, Ph. D.

Academic Editor

PLOS ONE

Journal Requirements:

Reviewers' comments:

Reviewer's Responses to Questions

**Comments to the Author**

1. Is the manuscript technically sound, and do the data support the conclusions?

Reviewer #1: Yes

Reviewer #2: Yes

2. Has the statistical analysis been performed appropriately and rigorously? 

Reviewer #1: Yes

Reviewer #2: Yes

3. Have the authors made all data underlying the findings in their manuscript fully available?

Reviewer #1: Yes

Reviewer #2: No

4. Is the manuscript presented in an intelligible fashion and written in standard English?

Reviewer #1: Yes

Reviewer #2: Yes

5. Review Comments to the Author

Reviewer #1: Review of McFarland et al. PLoS One 2020

General comments

This is an important subject, given the great decline of seafood in the ocean and the general suggestions to supplement or even replace wild seafood with aquacultured stocks that are often domestically bred. There is a non-genetic component (as has been emphasized in salmon) and a component of rearing (much better understood in terrestrial animal and plant breeding). The main limitation of this study is the possible conflation of the trait under selection (disease resistance) and correlated traits that are dragged along under selection, which in this study are unknown. The authors I am sure are aware of this. I wish marine workers would pay more attention to domestic plant and animal breeding studies as opposed to just the little we know about cultured marine fishes. I would refer the authors to specialist such as Phil Hedrick and Brian Charlesworth whose papers focused on population consequences of captive breeding and inbreeding depression – a principal danger found in domestic breeding. The authors allude to one important part of this in lines 88-90 but do not speak to inbreeding problem. And speaking of inbreeding: Another angle missed in this paper is the widespread study of genetic variation of marine species esp bivalves showing that they have even in wild populations, very reduced genetic variation relative to expectations from natural population sizes (see work of Hedgecock, Palumbi and others, see esp recent paper by Bernatchez et al 2018 (Evol. Applications) that relates replanting of hatchery reared oysters to seascape genetics in eastern Canada. This peculiar aspect of marine populations must also greatly influence the outcome of domestic rearing, since the source population may be genetically very peculiar, leading to uncertain outcomes. This only heightens how careful we have to think about the actual selective regime in a hatchery, even when you target a trait like disease resistance.

These larger considerations make me feel that the recommendations made by conservation organizations to use natural broodstock (eg discussed on lines 74-77) are very naïve (also we see these untested practices in coral replanting) and, frankly, have been untested as far as I know.

These comments do not invalidate the results, which seem to me to be sound and properly analyzed. But I am a bit skeptical that there are trait-specific generalizations to be had here without studying over multiple years and from multiple localities. These limitations are mentioned in this paper but perhaps can be amplified by the considerations mentioned in the first paragraph above.

Still some of the results are very useful, particularly the result that “Control larvae with ad libitum phytoplankton showed overall similar mortality and growth rates across all cohorts – there is a lesson here that is important. The authors mention the problem with using starvation (when does that happen in day and age of high nutrient input) but of course having the wrong phytoplankton food (given the prevalence of nuisance blooms these days) might be very common, even for larvae.

I am a little skeptical of the discussion concerning the potential role of DOM. Is DOC relevant without nitrogen and other essential element nutrients? Still the discussion is sound and mentions the limitation of conclusions. The authors might want to nevertheless look at a paper by Roditi et al. 2000 (Nature 407, 78–80, concerning the role of DOC in zebra mussels.

Methods: These seem to be sound and well within the recommendations of standard larval rearing techniques. Food is standard as well. Selection of stocks seems sensible. Respiration methods appropriate (maybe they should mention the minimum concentration reached in the use of the negative O2-time slope.

Inbreeding: I mentioned this above. The authors finally address this in discussion (533-537) since they do use a cross of two inbred lines. Not sure this means much unless you had direct genetic data, but I could be wrong. The paper they cite is based upon self-fertilized oysters so this may be an extreme case and not relevant here. Still..they do address this. They should discuss the inbred line issue in the introduction, in my view.

Overall I recommend publication with some attention paid to the general framework I mention. The experiments they do are well in line with past work and therefore specialists in this field will understand the significance and this is of general significance as well (as mentioned above).

Line by line edits:

64 organisms

267 ff error: is it 95pct confidence of mean? Or what?

389-390 My experience is with adults but replication of 2 I think is a real problem here.

Reviewer #2: T

The question raised by the authors is interesting and their experimental approach and data analyses for the most part fine. They could have perhaps evaluated growth rate in a more quantitative way, and although I don’t think it will change the results much, the manner in which they normalized oxygen consumption rate with shell length should be given more thought.

Abstract

A fascinating result was the rapid growth of the starved larvae once feeding was resumed so that by the end of the experiment, Day 29, there was very little difference in shell length among treatments (Fig. 4, S2 Fig). This result could be stressed more in the abstract (line 36). Oxygen consumption rates also recovered (Fig. 5) but there is no mention of the result in the abstract.

Introduction

The introduction provided good background for the study. Perhaps some minor re-organization connecting the hypothesis tested (64-66) and it’s basis (111-118). The sentence between 130 – 133 should be deleted as it is repetitive of the previous sentence.

Methods

I am not sure why S1 Fig is not included in the manuscript instead of in supplemental material. I found it useful for understanding the experimental design and suggest it be included in the text. I think in S1, however, there is a mislabel, shouldn’t Day 10 actually be Day 17? The starvation period was 10 days, and started on Day 7, correct?

Please explain how oxygen consumption rates were normalized by shell length (244). Although VO2 will scale to shell length, I would not expect it to be linear.

Results

Survival

Nice photos of the larval guts (e.g., S3 Fig), perhaps include one in the text.

Growth

Why were the actual growth rates of the larvae (i.e., �m d-1) not more quantitatively

compared, at least during certain time intervals. For example, once the refed larvae “recovered”, did they exhibit a significantly higher growth rate between days 22 and 29 than the control larvae?

I did not follow what was meant by “a steeper decline in size”, its sounds as if the larvae were shrinking (348-349).

(369-372) If shell length “showed no significant increase once food was removed” how was it AQF1/Fast exhibited a growth slow down? Also, what is being implied about the observation the starved and fed AQF1/Fast exhibited a slow down in shell growth during the same period. Perhaps just some rewriting here.

Respiration

In addition to my comment about VO2 normalization with shell length, the unit for respiration on Fig. 5 should be corrected. Also, it is unclear with the wild starved data why n = 2 on the first day and more replicates were run at later dates.

Discussion

430 – I do not understand “release from selection unlimited”.

6. PLOS authors have the option to publish the peer review history of their article (what does this mean?). If published, this will include your full peer review and any attached files.

Reviewer #1: No

Reviewer #2: No

---

## [Author Response · Author response to Decision Letter 0]

2 Jun 2020

Reviewers' comments:

Author’s response follows below each comment. 

Reviewer #1: Review of McFarland et al. PLoS One 2020

General comments

This is an important subject, given the great decline of seafood in the ocean and the general suggestions to supplement or even replace wild seafood with aquacultured stocks that are often domestically bred. There is a non-genetic component (as has been emphasized in salmon) and a component of rearing (much better understood in terrestrial animal and plant breeding). The main limitation of this study is the possible conflation of the trait under selection (disease resistance) and correlated traits that are dragged along under selection, which in this study are unknown. The authors I am sure are aware of this. I wish marine workers would pay more attention to domestic plant and animal breeding studies as opposed to just the little we know about cultured marine fishes. I would refer the authors to specialist such as – a principal danger found in domestic breeding. The authors allude to one important part of this in lines 88-90 but do not speak to inbreeding problem. And speaking of inbreeding: Another angle missed in this paper is the widespread study of genetic variation of marine species esp bivalves showing that they have even in wild populations, very reduced genetic variation relative to expectations from natural population sizes (see work of Hedgecock, Palumbi and others, see esp recent paper by Bernatchez et al 2018 (Evol. Applications) that relates replanting of hatchery reared oysters to seascape genetics in eastern Canada. This peculiar aspect of marine populations must also greatly influence the outcome of domestic rearing, since the source population may be genetically very peculiar, leading to uncertain outcomes. This only heightens how careful we have to think about the actual selective regime in a hatchery, even when you target a trait like disease resistance.

We added a brief mention of increased risk of using hatchery produced stocks due to increased potential for inbreeding and that this risk is increased due to the low number of effective breeders and potential for sweepstakes reproduction. Taking these concerns into account, our design reduced the potential for inbreeding in our experimental crosses between two genetically distinct selectively bred lines.

These larger considerations make me feel that the recommendations made by conservation organizations to use natural broodstock (eg discussed on lines 74-77) are very naïve (also we see these untested practices in coral replanting) and, frankly, have been untested as far as I know.

These comments do not invalidate the results, which seem to me to be sound and properly analyzed. But I am a bit skeptical that there are trait-specific generalizations to be had here without studying over multiple years and from multiple localities. These limitations are mentioned in this paper but perhaps can be amplified by the considerations mentioned in the first paragraph above.

Multiyear studies with multiple stocks (both wild and selected) would certainly improve our understanding of domestication effects. While it was outside the scope of this study, we will consider this in future studies. 

Still some of the results are very useful, particularly the result that “Control larvae with ad libitum phytoplankton showed overall similar mortality and growth rates across all cohorts – there is a lesson here that is important. The authors mention the problem with using starvation (when does that happen in day and age of high nutrient input) but of course having the wrong phytoplankton food (given the prevalence of nuisance blooms these days) might be very common, even for larvae.

I am a little skeptical of the discussion concerning the potential role of DOM. Is DOC relevant without nitrogen and other essential element nutrients? Still the discussion is sound and mentions the limitation of conclusions. The authors might want to nevertheless look at a paper by Roditi et al. 2000 (Nature 407, 78–80, concerning the role of DOC in zebra mussels. 

DOC was used as a proxy to estimate how much DOM our filtration system might be stripping from the natural river water. While we acknowledge there are many other factors involved in understanding the availability and utilization of micronutrients, the lack of a significant difference between raw river water and filtered water used in our experiments suggests that micronutrients may be similar to what would be observed in the natural environment. We appreciate the reference suggestion and have included it in our discussion as it helps support our discussion on micronutrient utilization. 

Methods: These seem to be sound and well within the recommendations of standard larval rearing techniques. Food is standard as well. Selection of stocks seems sensible. Respiration methods appropriate (maybe they should mention the minimum concentration reached in the use of the negative O2-time slope.

We have added details to the methods used for calculating the slope. 

Inbreeding: I mentioned this above. The authors finally address this in discussion (533-537) since they do use a cross of two inbred lines. Not sure this means much unless you had direct genetic data, but I could be wrong. The paper they cite is based upon self-fertilized oysters so this may be an extreme case and not relevant here. Still..they do address this. They should discuss the inbred line issue in the introduction, in my view.

A brief discussion of inbreeding was added to the introduction. 

Overall I recommend publication with some attention paid to the general framework I mention. The experiments they do are well in line with past work and therefore specialists in this field will understand the significance and this is of general significance as well (as mentioned above).

Line by line edits:

64 organisms - corrected

267 ff error: is it 95pct confidence of mean? Or what? 

Clarified with a general statement about results in the section ‘Statistical analysis’ 

389-390 My experience is with adults but replication of 2 I think is a real problem here. 

Yes, 2 is a problem, but the low replication was due to high mortality and low numbers of small / starving larvae. We chose to include this time point primarily for graphical observations suggesting low respiration during starvation. 

Reviewer #2: 

The question raised by the authors is interesting and their experimental approach and data analyses for the most part fine. They could have perhaps evaluated growth rate in a more quantitative way, and although I don’t think it will change the results much, the manner in which they normalized oxygen consumption rate with shell length should be given more thought. 

Yes, tissue weight would have been better than shell length, however we did not have the capabilities for such measurements. Given the concerns for using shell length and after a review of methods for reporting larval physiological rates in the relevant literature, we chose to use non-length standardize respiration rates. We would like to note that it did not significantly change our results and the general trends remain with both standardized and non-standardized respiration rates 

Abstract

A fascinating result was the rapid growth of the starved larvae once feeding was resumed so that by the end of the experiment, Day 29, there was very little difference in shell length among treatments (Fig. 4, S2 Fig). This result could be stressed more in the abstract (line 36). Oxygen consumption rates also recovered (Fig. 5) but there is no mention of the result in the abstract.

Restructured to abstract to include these results

Introduction

The introduction provided good background for the study. Perhaps some minor re-organization connecting the hypothesis tested (64-66) and it’s basis (111-118). The sentence between 130 – 133 should be deleted as it is repetitive of the previous sentence.

 Made the suggested rearrangements and deletions

Methods

I am not sure why S1 Fig is not included in the manuscript instead of in supplemental material. I found it useful for understanding the experimental design and suggest it be included in the text. I think in S1, however, there is a mislabel, shouldn’t Day 10 actually be Day 17? The starvation period was 10 days, and started on Day 7, correct? 

Corrected the mislabeled days in Fig S1 and included Fig. S1 in the main text

Please explain how oxygen consumption rates were normalized by shell length (244). Although VO2 will scale to shell length, I would not expect it to be linear. 

Given the concerns for using shell length and after a review of methods for reporting larval physiological rates in the relevant literature, we chose to use non-length standardize respiration rates. We would like to note that it did not significantly change our results and the general trends remain with both standardized and non-standardized respiration rates

Results

Survival

Nice photos of the larval guts (e.g., S3 Fig), perhaps include one in the text. 

We chose not to include the pictures for space. 

Growth

Why were the actual growth rates of the larvae (i.e., �m d-1) not more quantitatively

compared, at least during certain time intervals. For example, once the refed larvae “recovered”, did they exhibit a significantly higher growth rate between days 22 and 29 than the control larvae? 

We have included a table showing the growth rate during linear growth for each cohort as well as statistical comparison in the results. 

I did not follow what was meant by “a steeper decline in size”, its sounds as if the larvae were shrinking (348-349). 

This was in reference to the decrease in mean shell length due to settlement of larger individuals. We clarified this in the text.

(369-372) If shell length “showed no significant increase once food was removed” how was it AQF1/Fast exhibited a growth slow down? Also, what is being implied about the observation the starved and fed AQF1/Fast exhibited a slow down in shell growth during the same period. Perhaps just some rewriting here. 

This was reworded to improve clarity according to the reviewer’s suggestions

Respiration

In addition to my comment about VO2 normalization with shell length, the unit for respiration on Fig. 5 should be corrected. Also, it is unclear with the wild starved data why n = 2 on the first day and more replicates were run at later dates. 

Yes, 2 is a problem, but the low replication was due to high mortality and low numbers of small / starving larvae. We chose to include this time point primarily for graphical observations suggesting low respiration during starvation. This low N was the due to difficulty in obtaining enough larvae from each replicate to observe a signal in oxygen depletion. Unit for respiration was corrected in Fig. 5.

Discussion

430 – I do not understand “release from selection unlimited”. 

Corrected this sentence for clarification

---

## [Editor Report · Decision Letter 1]

4 Jun 2020

Are bivalves susceptible to domestication selection? Using starvation tolerance to test for potential trait changes in eastern oyster larvae

PONE-D-20-05015R1

Dear Dr. McFarland,

Thank your for submitting the revised version of your manuscript. I am satisfied with the changes made in response to the reviewers' concerns and suggestions.  I am  pleased to inform you that your manuscript has been judged scientifically suitable for publication and will be formally accepted for publication once it meets all outstanding technical requirements.

Kind regards,

Hans G. Dam, Ph. D.

Academic Editor

PLOS ONE
---

## [Editor Report · Acceptance letter]

17 Jun 2020

PONE-D-20-05015R1 

Are bivalves susceptible to domestication selection? Using starvation tolerance to test for potential trait changes in eastern oyster larvae 

Dear Dr. McFarland:

I'm pleased to inform you that your manuscript has been deemed suitable for publication in PLOS ONE. Congratulations! Your manuscript is now with our production department. 

Kind regards, 

on behalf of

Dr. Hans G. Dam 

Academic Editor

PLOS ONE